# Accelerating Learned Image Compression with Sensitivity-aware Embedding and Moving Average

## Abstract

As learned image compression (LIC) methods become increasingly computationally demanding, enhancing their training efficiency is crucial. This paper takes a step forward in accelerating the training of LIC methods by modeling the training dynamics. We first propose a Sensitivity-aware True and Dummy Embedding Training mechanism (STDET) that clusters LIC model parameters into few separate modes where parameters are expressed as affine transformations of reference parameters within the same mode. By further utilizing the stable intra-mode correlations throughout training and parameter sensitivities, we gradually embed non-reference parameters, reducing the number of trainable parameters. Additionally, we incorporate a Sampling-then-Moving Average (SMA) technique, interpolating sampled weights from stochastic gradient descent (SGD) training to obtain the moving averaged weights, ensuring smooth temporal behavior and minimizing training state variances. Overall, our method significantly reduces training space dimensions and the number of trainable parameters without sacrificing model performance, thus accelerating model convergence. We also provide a theoretical analysis of the Noisy quadratic model, showing that the proposed method achieves a lower training variance than standard SGD. Our approach offers valuable insights for further developing efficient training methods for LICs. The code will be publicly available.

## 1 Introduction

With the widespread adoption of high-resolution cameras and the increasing prevalence of image-centric social media platforms and digital galleries, images have become a predominant form of media in daily life. The large file sizes of these high-resolution images place considerable demands on both transmission bandwidth and storage capacity. To address these challenges, lossy image compression has emerged as a critical technique for achieving efficient visual communication.

Recently, learned image compression (LIC) methods have garnered significant attention due to their remarkable performance (He et al., 2022; Liu et al., 2023; Li et al., 2024a). Despite these advancements, LICs currently face a core challenge: high training complexity. Designing a new method requires a substantial amount of computational resources, which severely hinders the emergence of new approaches. Modern LICs typically feature a vast number of parameters that, while enhancing performance, also introduce critical issues such as an overwhelming computational burden and prolonged training times. For instance, training the FLIC models (Li et al., 2024a) takes up to 520 hours (21 days) using a single 4090 GPU (Tab. 1), while the design process of the FLIC models likely requires even more GPU resources, as it involves extensive experimentation, including hyper-parameter tuning, structural adjustments, and other iterative processes. If the training speed of these models is not improved, the time required to develop new methods may become prohibitive. Therefore, developing strategies to reduce training times is of paramount importance.

While there have been efforts to design more efficient LIC methods (e.g., (He et al., 2022; Zhang et al., 2024b)), improving the training efficiency of LIC methods has not received significant attention. A promising and effective direction is to train LICs within low-dimensional subspaces as been examined in the context of image classification tasks (Li et al., 2018; Gressmann et al., 2020; Li et al., 2022a;b; Brokman et al., 2024).

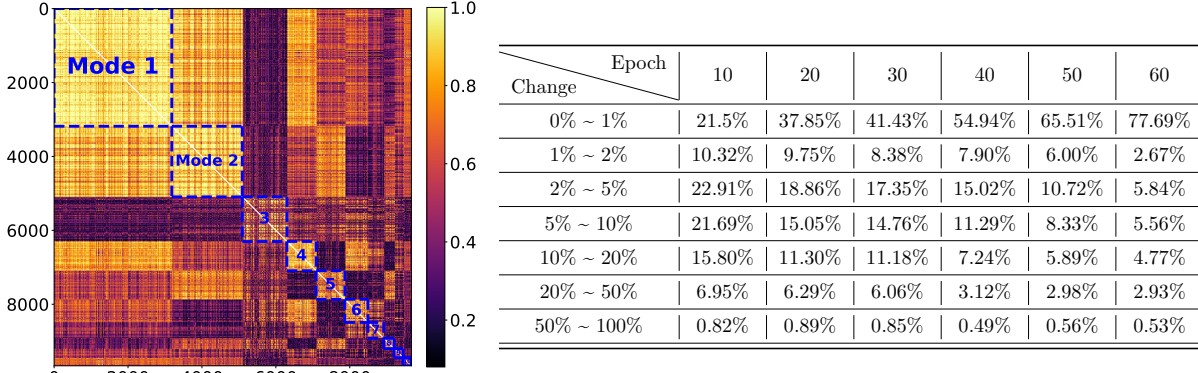

Figure 1: ELIC (He et al., 2022) model, $\lambda = 0.0018$. (a) Clustered correlation matrix of sampled 10k parameter trajectories trained on COCO2017 dataset (Lin et al., 2014), decomposed to 10 modes. The diagonal block structure indicates high correlations of the parameters within each mode, which shows the accurate representation of the proposed method. (b) The table shows the percentage of affine coefficients $\{k_i\}_{i=1}^N$ relative to the total number of coefficients, grouped by relative change intervals, at different epochs during the training of the ELIC model. These relative changes are measured against the final coefficient values at epoch 70. The rows correspond to the relative change intervals (0% - 1%, 1% - 2%, ..., 50% - 100%), indicating how much the coefficients have relatively changed compared to their values at epoch 70. The columns represent specific epochs (10, 20, 30, 40, 50, 60). The percentages indicate the proportion of coefficients that fall within each relative change interval at the corresponding epoch. The table reveals that most coefficients either remain stable or undergo only minor changes as training progresses from epoch 10 to 60. Notably, the proportion of coefficients in the 0% to 1% interval increases significantly from 21.5% at epoch 10 to 77.69% at epoch 60, indicating a marked stabilization of the affine coefficients over time.

Instead of exploiting the full parameter space, which can be very large (in millions or even billions), subspace training constrains the training trajectory to a low-dimensional subspace. These approaches are based on the hypothesis that "learned over-parameterized models reside in a low intrinsic dimension" (Li et al., 2018; Aghajanyan et al., 2021). In such subspaces, the degree of freedom for training is substantially reduced (to dozens or hundreds), leading to many favorable properties, such as fast convergence (Li et al., 2023a), robust performance (Li et al., 2022a), and theoretical insights (Li et al., 2018; 2022a).

In this work, we present an innovative approach to improve training efficiency by modeling training dynamics in the reduced dimensional space. Our proposed method draws inspiration from Dynamic Mode Decomposition (DMD) (Schmid, 2010; 2022; Brokman et al., 2024; Mudrik et al., 2024), a technique traditionally used in fluid dynamics to decompose complex systems into simpler modes. We extend this concept to LIC by proposing the Sensitivity-aware True and Dummy Embedding Training (STDET) mechanism. The fundamental principle of our approach is the recognition that LIC parameters are highly correlated and can be effectively represented by few distinct "Modes" (see Fig. 1), which capture their intrinsic dimensions. Within each mode, the parameters are aptly modeled through an affine transformation of the reference parameters, given their significant correlation. We also observe that after the initial head-stage training, the relative changes of the affine coefficients become negligible and the coefficients tend to remain stable throughout the training, as illustrated in Fig. 1. It is evident that the majority of the coefficients exhibit 0% to 1% relative changes, as indicated by the blue line, with the percentage in this interval continuing to increase. This stability permits a focused update of reference parameters and then allows for the "embedding" of non-reference parameters that have relatively invariant coefficients. Once embedded, these parameters are no longer trainable. Subsequently, updates of these embedded parameters are done solely through the fixed coefficients affine transformation of the reference parameters. Non-embedded and reference parameters are still updated via normal training. This mechanism significantly reduces the number of training parameters and the dimension of LIC models in practice along the training period, thereby expediting convergence, as demonstrated in Fig. 2. Ideally, the training parameters and dimensions could be reduced to the number of "Modes". Additionally, to ensure smooth temporal behaviors and minimize training variances which is

the core requirement of STDET, we introduce the well-known moving average techniques to the training phase. Our proposed Sampling-then-Moving Average (SMA) method interpolates the periodically sampled parameter states from SGD training to derive moving averaged parameters, thus enhancing stability and reducing the variance of final model states. Our method is substantiated by comprehensive experiments on various complex LIC models and comparisons with other efficient training methods. The superiority of the proposed method is also validated through theoretical analysis on the noisy quadratic model.

Our contributions are summarized as follows:

- We propose the Sensitivity-aware True and Dummy Embedding Training (STDET) mechanism, which approximates the SGD training of LICs in a low-dimensional space (Sec. 3.2).

- We introduce Sampling-then-Moving Average (SMA) to ensure the smooth temporal behavior required by STDET, reducing the variances of final states and enhancing training stability (Sec. 3.3).

- We provide a theoretical analysis using the noisy quadratic model to demonstrate the low training variance of the proposed method (Sec. A.6).

- Overall, our proposed method significantly accelerates the training of LICs while reducing the number of trainable parameters and dimensions without compromising performance (Sec. 4.2, Sec. 4.3).

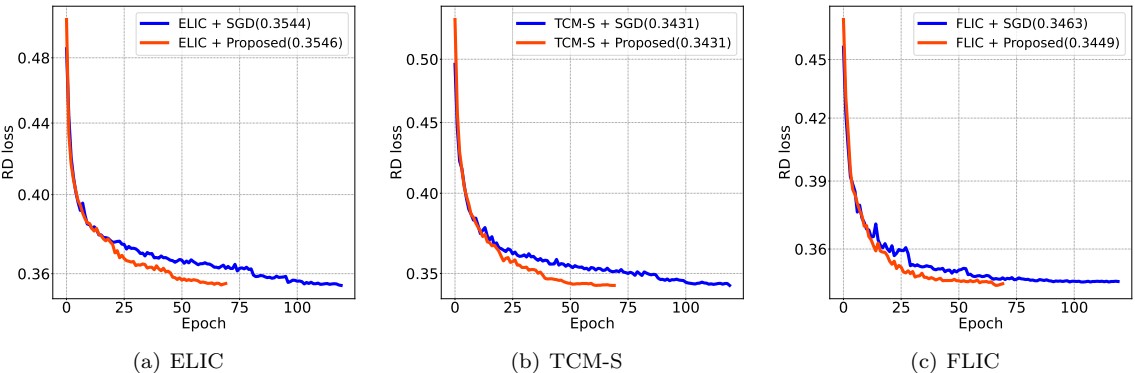

(a) ELIC             (b) TCM-S             (c) FLIC

Figure 2: **Testing loss comparison of various methods.** *Please zoom in for more details.* The proposed method clearly converges much faster than standard SGD on various LICs. Additionally, as shown in the upper right corner, our method achieves a similar final convergence compared to SGD. $\lambda = 0.0018$, Testing RD loss $= \lambda \cdot 255^2 \cdot \text{MSE} + \text{Bpp}$. Evaluated on Kodak dataset.

## 2 Related work

### 2.1 Efficient learned image compression

To address the issue of high computational complexity in learned image compression, various approaches have been proposed: Minnen & Singh (2020) develop a channel-wise autoregressive model to capture channel relationships instead of spatial ones. He et al. (2021) propose a parallelizable two-pass checkerboard model to accelerate spatial autoregressive models. Tao et al. (2023) introduce dynamic transform routing to activate optimally-sized sub-CAEs within a slimmable supernet, conserving computational resources. Duan et al. (2023) incorporate a hierarchical VAE for probabilistic modeling to generalize autoregressive models. Guo-Hua et al. (2023) propose an efficient single-model variable-bit-rate codec capable of running at 30 FPS with 768x512 input images. Yang & Mandt (2023) adopt shallow or linear decoding transforms to reduce decoding complexity. Ali et al. (2023) introduce a correlation loss that forces the latents to be spatially decorrelated, fitting them to an independent probability model and eliminating the need for autoregressive models. Minnen & Johnston (2023) conduct a rate-distortion-computation study, leading to a family of model architectures that optimize the trade-off between computational requirements and R-D performance. Zhang et al. (2024a) propose contextual clustering to replace computationally intensive self-attention mechanisms. Zhang et al.

(2024b) test various transforms and context models to identify a series of efficient LIC models. However, these works primarily focus on test-time efficiency and often overlook the significant computational resources consumed during the training phase. Training complex LICs is resource-intensive and optimizing training efficiency is crucial.

## 2.2 Low dimension training/finetuning of modern neural networks

Training or fine-tuning neural networks in low-dimensional spaces has garnered significant attention recently. Li et al. (2018) propose training neural networks in random subspaces of the parameter space to identify the minimum dimension required for effective solutions. Gressmann et al. (2020) enhance training performance in random bases by considering layerwise structures and redrawing the random bases iteratively. Li et al. (2022a) utilizes top eigenvectors of the covariance matrix to span the training space. Additionally, Li et al. (2022b) apply subspace training to adversarial training, mitigating catastrophic and robust overfitting. Further advancements include Li et al. (2023a), which propose trainable weight averaging to optimize historical solutions in the reduced subspace, and Aghajanyan et al. (2021), which demonstrate that learned over-parameterized models reside in a low intrinsic dimension. Hu et al. (2022) introduce Low-Rank Adaptation, based on the premise that weight changes during model adaptation have a low "intrinsic rank". Similarly Ding et al. (2023) present a delta-tuning method optimizing only a small portion of model parameters, while Jia et al. (2022) propose Visual Prompt Tuning, introducing less than 1% of trainable parameters in the input space. Lastly, Barbano et al. (2024) constrain deep image prior optimization to a sparse linear subspace of parameters, employing a synergy of dimensionality reduction techniques and second-order optimization methods. These works highlight the elegant performance of the low-dimensional hypothesis, which has not yet been thoroughly explored in the context of learned image compression.

# 3 Proposed method

## 3.1 Preliminaries on learned image compression

Currently, LICs predominantly utilize the non-linear transform coding framework (Ballé et al., 2020), where the transform function is parameterized by neural networks. This framework encodes data into a discrete representation for de-correlation and energy compression, and then an entropy model is used to estimate the probability distribution of the discrete representation for entropy coding, as illustrated in Fig. 3. The process begins with an autoencoder applying a learned nonlinear analysis transform, $g_a$, to map the input image $x$ to a lower-dimension latent variable $y = g_a(x)$. This latent variable $y$ is then quantized to $\hat{y}$ using a quantization function $Q$ and encoded into a bitstream via a lossless codec (Duda, 2009), resulting in a much smaller file size. The decoder then reads the bitstream of $\hat{y}$, applies the synthesis transform $g_s$, and reconstructs the image as $\hat{x}$, with the goal of minimizing the distortion $\mathcal{D}(x, \hat{x})$. Minimizing the entropy of $\hat{y}$ involves learning its probability distribution through entropy modeling, which is achieved using an entropy model $P$ that includes both forward and backward adaptation methods. The forward adaptation employs a hyperprior estimator, which is another autoencoder with its own hyper analysis transform $h_a$ and hyper synthesis transform $h_s$. This hyperprior effectively captures spatial dependencies in the latent representation $y$ and generates a separately encoded latent variable $\hat{z}$, which is sent to the decoder. A factorized density model (Ballé et al., 2018) is used to learn local histograms, estimating the probability mass $p_{\hat{z}}(\hat{z}|\psi_f)$ with model parameters $\psi_f$. The backward adaptation estimates the entropy parameters of the current latent element $\hat{y}_i$ based on previously coded elements $\hat{y}_{<i}$, where $i$ is the latent element index, exploring the redundancy between latent elements. This is achieved through a network $g_b$ that operates in an autoregressive manner over the spatial dimension, channel dimension, or a combination of both. The outputs from $g_b$ and the hyperprior are then used to parameterize the conditional distribution $p_{\hat{y}}(\hat{y}|\hat{z})$ via an entropy parameters network $g_e$. The LIC framework can be formulated as:

$$\begin{aligned}
\hat{y} &= Q(g_a(x; \phi_a)), \\
\hat{x} &= g_s(\hat{y}; \phi_s), \\
\hat{z} &= Q(h_a(y; \theta_a)), \\
p_{\hat{y}_i}(\hat{y}_i|\hat{z}) &\leftarrow g_e(g_b(\hat{y}_{<i}; \theta_b), h_s(\hat{z}; \theta_s); \psi_e),
\end{aligned} \tag{1}$$

with the Lagrange multiplier-based rate-distortion (R-D) loss function $\mathcal{L}$ for end-to-end training:

$$\begin{aligned}
\mathcal{L}(\phi, \theta, \psi) &= \mathcal{R}(\hat{y}) + \mathcal{R}(\hat{z}) + \lambda \cdot \mathcal{D}(x, \hat{x}), \\
&= \mathbb{E}[\log_2(p_{\hat{y}}(\hat{y}|\hat{z}))] + \mathbb{E}[\log_2(p_{\hat{z}}(\hat{z}|\psi))] + \lambda \cdot \mathcal{D}(x, \hat{x}),
\end{aligned} \tag{2}$$

where $\mathcal{D}(x, \hat{x})$ measures the distortion (e.g. MSE) between the original image $x$ and the reconstructed image $\hat{x}$. $\mathcal{R}(\hat{z})$ and $\mathcal{R}(\hat{y})$ represent the rate of $\hat{z}$ and $\hat{y}$, respectively.

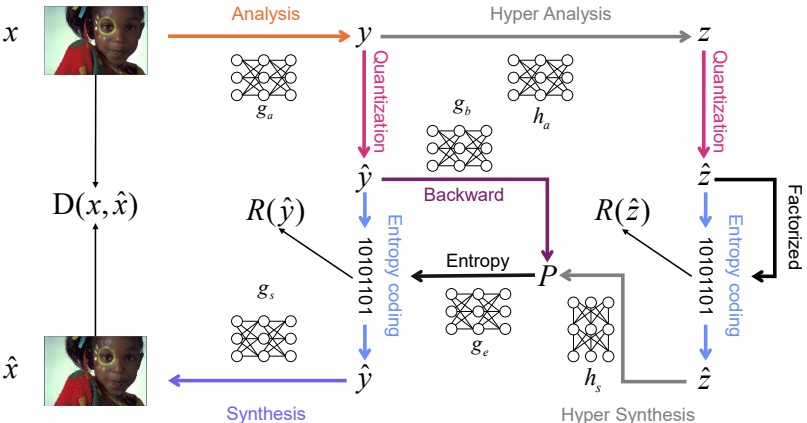

Figure 3: A typical pipeline in learned image compression.

## 3.2 Sensitivity-aware True and Dummy Embedding Training Mechanism

### 3.2.1 Linear representations of non-linear dynamical systems:

As shown in Sec. 3.1 and Fig. 3, LICs heavily rely on neural networks for transform functions and distribution parameters estimation, which are inherently nonlinear and complex systems. However, recent studies (Lusch et al., 2018; Razzhigaev et al., 2024) demonstrate that, under certain conditions, neural networks exhibit intrinsic linear behavior. This linearity can be effectively harnessed using techniques such as Dynamic Mode Decomposition (DMD) (Schmid, 2010; 2022; Brokman et al., 2024; Mudrik et al., 2024) and Koopman theory (Dogra & Redman, 2020; Brunton et al., 2022), which provide linear approximations of the network's dynamics. By leveraging these methods, the analysis and efficiency of various tasks can be significantly improved. Approximating nonlinear LICs with linear operations could offer numerous benefits, including faster convergence, better interpretability, and greater stability. Therefore, there is substantial potential in employing linear approximations for these complex LICs to achieve enhanced efficiency.

We select a variant of DMD as the core technique for the proposed method due to its simpler and more intuitive representation compared to Koopman theory methods. Let $\mathbf{W} = \{w_i(t)\}_{i=1}^{N} \cup \{w_m(t)\}_{m=1}^{M}$ denote the set of LIC's trainable parameters, where $i$ is the index of the non-reference parameters, m is the index of reference parameters, $t$ represents the training time, and $N$ is the total number of parameters. We refer to $\mathbf{W}$ as the set of 'trajectories,' with each $w_i(t)/w_m(t)$ representing an individual 'trajectory.' DMD methods aim to represent the dynamics of complex models by decomposing them into modes, where each mode is captured by a concise set of reference trajectories $w_m(t)$. This can be expressed as: $w_i(t) \approx \sum_m k_{i,m} w_m(t) + d_i$, where $k_{i,m}$ and $d_i$ are scalars. Specially, we employ Correlation Mode Decomposition (CMD) (Brokman et al., 2024), which further simplifies this representation:

$$w_i(t) \approx k_i w_m(t) + d_i, \tag{3}$$

where $k_i$ and $d_i$ are scalar affine coefficients associated with $w_i$. CMD operates under the assumption that a complex system can be represented by several modes, and within each mode, an individual parameter can be effectively represented by the affine transformation of the mode. This assumption also holds true in the training of LICs, as shown in Fig. 1. CMD can be seen as a special, simplified case of DMDs, where the theoretical dimension corresponds to the number of reference trajectories. With CMD, we achieve a linear approximation of complex nonlinear LICs, which present desirable features.

**Notation:** Let's consider a discrete-time setting in the training process, $t \in [1, \ldots, E]$, where $E$ is the number of epochs. Let $N_m$ be the number of trajectories in mode $m$. Then each parameter $w_m, w_i \in \mathbb{R}^E$, the parameter set $\mathbf{W} \in \mathbb{R}^{N \times E}$, the mode $m$ parameter set $\mathbf{W}_m \in \mathbb{R}^{N_m \times E}$. We use $(t)$ represent at time $t$, and $_t$ represent up to time $t$. We consider 1D vectors such as $u \in \mathbb{R}^E$ to be row vectors, and define the centering operation $\bar{u} := u - \frac{1}{E} \sum_{t=1}^{E} u(t)$, outer product $\langle u, v \rangle = uv^T$, euclidean norm $\|u\| = \sqrt{uu^T}$, and $\text{corr}(u, v) = \frac{\langle \bar{u}, \bar{v} \rangle}{\|\bar{u}\|\|\bar{v}\|}$. Let $K, D \in \mathbb{R}^N$ such that $K(i) = k_i$, $D(i) = d_i$, and let $K_m, D_m \in \mathbb{R}^{N_m}$ be the parts of $K, D$ corresponding to the parameters in mode $m$. Denote matrix form $\tilde{w}_m = \begin{bmatrix} w_m \\ \vec{1} \end{bmatrix} \in \mathbb{R}^{2 \times E}$, where $\vec{1} = (1, 1, \ldots, 1) \in \mathbb{R}^E$, and $\tilde{K}_m = [K_m^T, D_m^T] \in \mathbb{R}^{N_m \times 2}$. In this context, Eq. 3 reads as:

$$\mathbf{W}_m \approx \tilde{K}_m \tilde{w}_m. \tag{4}$$

We then use Eq. 4 and these notations to model the LICs trainable parameters. To clarify the term in the following section, we have two types of parameters: reference and non-reference. Non-reference parameters are further divided into embeddable and non-embeddable.

### 3.2.2 Proposed STDET:

The proposed STDET mechanism builds on the notation and concepts defined above. We perform CMD after an initial head-stage training to identify the reference parameters and modes. We then estimate the sensitivity of the LIC parameters to determine which parameters are "embeddable". Next, The proposed STDET iteratively updates the affine coefficients based on the updated neural state. Subsequently, we embed the non-reference parameters every few epochs according to the stability of the coefficients and the parameter sensitivity. Our method reduces both the training space dimension and the number of trainable parameters as training moves forward, dynamically optimizing the overall training process.

**Step 1: Mode decomposition.** We select reference parameters after the initial head-stage training. The LIC is trained normally for a predefined number of epochs $F$ to obtain the trajectories $w_{i,F} \in \mathbf{W}_F$ up to epoch $F$. We then perform Correlation Mode Decomposition (CMD) (Brokman et al., 2024) to get the reference parameters $w_{m,F}$, the mode of each $w_{i,F} \in \mathbf{W}_F$, and the affine coefficients $\tilde{K}_m(F)$ at epoch $F$ based on Eq. 4. This CMD determination and association is performed only once. Once $w_{i,F}$ is associated with mode $m$ and reference parameter $w_{m,F}$, $w_i$ remains in mode $m$ with reference parameter $w_m$ for the entire training process.

**Step 2: Parameter Sensitivity Estimation.** Before embedding the non-reference parameters in LIC models, it is essential to determine which parameters are "embeddable" as some parameters may be extremely sensitive to perturbations, leading to significant performance drops (Weng et al., 2020). We do not embed these parameters. Accurately assessing parameter sensitivity is challenging due to the complexity involved in individually perturbing each parameter and measuring the prediction error. Given the lack of generally reliable estimation methods (Yvinec et al., 2023), we employ a combination of accurate layer-wise assessment and rough parameter-wise estimation to rank the sensitivity of the LIC parameters.

We begin by individually perturbing each layer and measuring the impact on the rate-distortion (R-D) loss using a randomly sampled portion of the training dataset $\mathcal{X} = (x_1, \cdots, x_n)$ (with $n = 256$ in practice). Different strategies are employed for evaluating analysis transform $g_a$, synthesis transform $g_s$ and hyper analysis transform $h_a$, hyper synthesis transform $h_s$, backward network $g_b$, entropy parameters network $g_e$ based on their unique behaviors. For layers in $g_a, g_s$, we compute the R-D loss before and after the perturbation. In contrast, for layers in $h_a, h_s, g_b, g_e$, the perturbation does not affect the reconstruction performance (the $D$ term), as these networks only involve calculating the $R$ term, as shown in Fig. 3. Thus, we only need a single forward pass to obtain $\hat{y}$, and then we individually perturb $h_a, h_s, g_b, g_e$ layers based on the existing $\hat{y}$. This approach allows us to avoid recurrently evaluating $\hat{y}$ and focus solely on changes in $R$, thereby further reducing complexity. For the perturbation, Gaussian noise sampled from $N(0, \sigma^2)$ is added to all parameters when perturbing each layer, where $\sigma$ is a fraction of the maximum parameter value in the layer. After calculating the R-D loss (or $R$ term) increase for all layers, those showing significant increases in R-D loss (or $R$ term) upon perturbation are deemed more sensitive. Following Novak et al. (2018) and Yang et al. (2023), which demonstrate that approximately 75% of parameters can be pruned without significantly

affecting performance, we infer that only 25% of the parameters are sensitive. To identify these sensitive parameters, we employ a straightforward and intuitive half-half strategy. We first identify the top 50% most sensitive layers, with 25% from $g_a$ and $g_s$, and 25% from $h_a$, $h_s$, $g_c$, and $g_e$.

Afterward, we estimate the parameter-wise sensitivity for the top 50% sensitive layers' parameters using the first-order estimation (Molchanov et al., 2019):

$$|\mathbf{W}| \odot \nabla : (f, \mathcal{X}) \rightarrow \left( \mathbb{E}_{\mathcal{X}} \left[ |w_i| \cdot \frac{\partial f}{\partial w_i} \right] \right)_{i \in \{1, \ldots, N\}}, \tag{5}$$

where $\odot$ denotes elementwise multiplication. This function combines both the magnitude of the parameter absolute value $|w_i|$ and the gradients of the R-D function $f$ w.r.t. each parameter $\frac{\partial f}{\partial w_i}$, which is effective in practice (Yvinec et al., 2023). By utilizing the estimated rough parameter-wise sensitivity, we identify the 50% relative sensitive parameters from these layers and consider these parameters not "embeddable" (25% of total parameters), whereas the remaining 75% of parameters are "embeddable." We calculate the sensitivity only once after the head-stage training.

**Step 3. Affine coefficients update.** With the affine coefficients obtained in step 1 at epoch $F$, we now need to update them for each new epoch to observe their temporal behaviors. Let $\tilde{K}_m(t)$ be coefficients $\tilde{K}_m$ evaluated at time $t$. Given the previous epoch $\tilde{K}_m(t-1)$, we can update $\tilde{K}_m$ using Eq. 6:

$$\begin{aligned} \tilde{K}_m(t) = &\left( \tilde{K}_m(t-1)(\tilde{w}_{m,t-1}\tilde{w}_{m,t-1}^T) + W_m(t)\tilde{w}_m^T(t) \right) \\ &\times (\tilde{w}_{m,t-1}\tilde{w}_{m,t-1}^T + \begin{pmatrix} w_m^2(t) & w_m(t) \\ w_m(t) & 1 \end{pmatrix})^{-1}, \end{aligned} \tag{6}$$

where $W_m(t)$ and $\tilde{w}_m(t)$ are the $t$-th columns of $W_m$ and $\tilde{w}_m$, respectively, and $\tilde{w}_{m,t}$ comprises columns 1 through $t$ of $\tilde{w}_m$. Eq. 6 is used for iterative updates after each new epoch.

**Step 4. True and Dummy Embedding.** As we update the coefficients at each new epoch, it is observed that the affine coefficients $k_i$ and $d_i$ tend to stabilize after the head-stage training, as shown in Fig. 1. We can therefore fix $k_i$ and $d_i$ based on their long-term changes $c_i$, evaluated every $L$ epochs. $c_i(t)$ is defined as the Euclidean distance between the current values of $k_i$ and $d_i$ and their $L$ epochs earlier values:

$$c_i(t) = \sqrt{\|k_i(t) - k_i(t-L)\|^2 + \|d_i(t) - d_i(t-L)\|^2}. \tag{7}$$

For $k_i$ and $d_i$ associated with "embeddable" parameters, those with the least $P\%$ changes in terms of $c_i$ are considered embedded coefficients, meaning they are fixed from this point onward. The corresponding $w_i$ are the true embedded non-reference parameters. Note that we do not embed the reference parameters $\{w_m\}_{m=1}^M$. While the embedded parameters are no longer trainable, they still evolve as $w_m(t)$ changes, following $w_i = k_i w_m(t) + d_i$. Consequently, the dimension of the training space and the number of trainable parameters gradually reduce as parameters are embedded. This reduction in the degrees of freedom in the training dynamics facilitates the convergence of the training process. Ultimately, the dimension of the final state training space and the number of trainable parameters converge to the number of modes $M$ in the theoretical case. In practice, we typically embed 50% of the parameters in our experiments.

Additionally, we reinitialize additional $\frac{tP}{2}\%$ parameters that have the least $c_i$ changes excluding the truly embedded parameters. We assign these parameters the values from their embedding versions: $w_i \leftarrow k_i w_m(t) + d_i$. We refer to this process as dummy embedding. In the next epoch, the dummy embedded parameters are updated like the non-embedded parameters. Empirical results indicate that dummy embedding enhances performance. One explanation is that this dummy embedding scheme simulates the Random Weight Perturbation (RWP) (Li et al., 2024b; Kanashiro Pereira et al., 2021), which has been shown to improve performance and generalization. Throughout the entire training process, we execute Step 1 and Step 2 once after the predefined epoch $F$. Subsequently, we repeatedly perform Step 3 and Step 4 at each new training epoch to embed parameters. The complete STDET algorithm is detailed in the Appendix in Algorithm 1.

### 3.3 Sampling-then-Moving Average

#### 3.3.1 Moving average:

As described in Sec. 3.2, the proposed STDET relies heavily on stable correlation and smooth temporal behavior when embedding parameters, necessitating reduced training variance. However, due to the inherent complexity of training LIC models, maintaining such stability can be challenging. To regulate this, we incorporate the moving average technique directly into the training phase, which helps to smooth out noise and randomness to ensure the consistency and stability of the training parameters (Martens, 2020; Chen et al., 2021; Morales-Brotons et al., 2024).

The moving average technique is a straightforward and efficient method for consolidating multiple model checkpoints into a single one, thereby enhancing performance, robustness, and generalization (Li et al., 2023b). It is particularly effective in models that exhibit a degree of similarity across checkpoints and does not introduce additional computational overhead. Among the most commonly used techniques is the Exponential Moving Average (EMA) (Szegedy et al., 2016; Polyak & Juditsky, 1992; He et al., 2020).

EMA integrates early-stage states while assigning higher importance to more recent ones, which allows the model to adapt more quickly to changes during training. The EMA parameters are calculated as follows:

$$w_{\text{EMA}}(0) = w(0), \ w_{\text{EMA}}(t+1) = (1-\alpha)w_{\text{EMA}}(t) + \alpha w(t+1), \tag{8}$$

where $\alpha$ is a moving average factor that determines the weight given to recent versus older states.

Moving Average is argued to find flatter solutions in the loss asymmetric valleys than SGD, thus generalizing better to unseen data with a potential explanation that the loss function near a minimum is often asymmetric, sharp in some directions, and flat in others. While SGD tends to land near a sharp ascent, averaging iterates biased solutions towards a flat region (He et al., 2019).

#### 3.3.2 Proposed SMA:

While EMA is designed for the testing phase, where its parameters do not influence the training process and are only used as the final model checkpoint during testing, we propose integrating EMA into the training phase itself to achieve a smoother training trajectory.

Specifically, we sample states from the SGD trajectories at regular intervals $l$ and use the moving average rule to update the training parameters. Following EMA principles, the proposed Sampling-then-Moving Average (SMA) maintains a set of SGD parameters $w$ and a set of SMA parameters $w_{\text{SMA}}$. The SGD parameters are obtained by applying the SGD optimization algorithm to batches of training examples. After every $l$ optimizer updates using SGD, we sample the current states and update the SMA parameters through linear interpolation based on EMA rules as in Eq. 8. Each time the SGD parameters are sampled and the SMA parameters are updated, we synchronize the SMA parameters to the SGD parameters to give a new starting point. The trajectory of the SMA parameters $w_{\text{SMA}}$ is thus characterized as an exponential moving average of the sampled SGD parameters states $w_l$. After $l$ optimizer steps, we have:

$$
\begin{aligned}
w_{\text{SMA}}(t+1) &= (1-\alpha)w_{\text{SMA}}(t) + \alpha w_l(t) \\
&= \alpha\big[w_l(t) + (1-\alpha)w_l(t-1) + \cdots \\
&\quad + (1-\alpha)^{t-1}w_l(0)\big] + (1-\alpha)^t w_{\text{SMA}}(0),
\end{aligned}
\tag{9}
$$

where $\alpha$ is the moving average factor. The SMA parameters leverage recent states from SGD optimization while retaining some influence from earlier SGD parameters to effectively regulate temporal behavior and reduce variance. The additional moving average update and synchronization introduce negligible complexity. We further reduce the complexity by treating the sampling and updating as a single optimizer step, ensuring that the total number of iterations within each epoch remains unchanged.

By jointly applying STDET and SMA, the training trajectories of non-reference parameters $w_i$ are modeled as follows:

$$
w_i(t) \leftarrow \begin{cases} k_i w_m(t) + d_i & \text{if } w_i(t) \text{ is embedded} \\ \text{SMA update} & \text{else} \end{cases}
\tag{10}
$$

# 4 Experimental results

## 4.1 Experimental settings

**Training.** We use the COCO2017 dataset (Lin et al., 2014) for training, which contains 118,287 images, each having around $640 \times 420$ pixels. We randomly crop $256 \times 256$ patches from these images. All models are trained using the Lagrange multiplier-based rate-distortion loss as defined in Eq. 2.

Following the settings of CompressAI (Bégaint et al., 2020), we set $\lambda$ to $\{18, 35, 67, 130, 250, 483\} \times 10^{-4}$. For all models in the "+ SGD" series, we train each model using the Adam optimizer[1] with $\beta_1 = 0.9$ and $\beta_2 = 0.999$. The $\lambda = 0.0018$ models are trained for 120 epochs. For models with other $\lambda$ values, we fine-tune the model trained with $\lambda = 0.0018$ for an additional 80 epochs. For all models in the "+ Proposed" series, we train each $\lambda = 0.0018$ model using the Adam optimizer for 70 epochs. For models with other $\lambda$ values, we fine-tune the model trained with $\lambda = 0.0018$ for an additional 50 epochs. More details are provided in the Appendix A.1.

**Testing.** Three widely used benchmark datasets, including Kodak (Franzen), Tecnick (Asuni et al., 2014), and CLIC 2022 (CLIC), are used to evaluate the performance of the proposed method.

## 4.2 Quantitative results

We compare our proposed method with standard SGD-trained models on prevalent complex LICs including ELIC (He et al., 2022), TCM-S (Liu et al., 2023), and FLIC (Li et al., 2024a) to demonstrate its superior performance. We use SGD-trained models as the anchor to calculate BD-Rate (Bjøntegaard, 2001).

Tab. 1 presents the BD-Rate reduction of the proposed method compared to the SGD anchors across three datasets. Our proposed method consistently achieves comparable final results to all methods across these datasets, which include both normal and high-resolution images, demonstrating its effectiveness. For example, on the Kodak dataset, "ELIC + Proposed" achieves -0.69% against "ELIC + SGD". Fig. 4 further illustrates the R-D curves of all methods, showing that the R-D points of the SGD-trained model and the model trained by the proposed method almost overlap.

Table 1: Computational Complexity and BD-Rate Compared to SGD

| Method | | Training time ↓ | Total trainable params ↓ | Final trainable params ↓ | BD-Rate (%) ↓ | | |
| --- | --- | --- | --- | --- | --- | --- | --- |
| | | | | | Kodak | Tecnick | CLIC2022 |
| ELIC (He et al., 2022) | + SGD | 165h | 18,418M | 35.42M | 0% | 0% | 0% |
| | + Proposed | **101h** | **9,519M** | **20.66M** | **-0.69%** | **-0.68%** | **-0.51%** |
| TCM-S (Liu et al., 2023) | + SGD | 346h | 23,493M | 45.18M | 0% | 0% | 0% |
| | + Proposed | **213h** | **12,142M** | **26.34M** | **-0.22%** | **-0.39%** | **-0.54%** |
| FLIC (Li et al., 2024a) | + SGD | 520h | 36,899M | 70.96M | 0% | 0% | 0% |
| | + Proposed | **320h** | **19,070M** | **41.39M** | **-1.30%** | **-1.22%** | **-1.42%** |

**Training Conditions**:$1 \times$ Nvidia 4090 GPU, i9-14900K CPU, 128GB RAM. **Bold** represents better performance.

We also compare our approach with other efficient training methods, including P-SGD (Li et al., 2022a), P-BFGS (Li et al., 2022a), and TWA (Li et al., 2023a). P-SGD employs SGD in a projected subspace derived from PCA. P-BFGS uses the quasi-Newton method BFGS within the same projected subspace. TWA utilizes Schmidt orthogonalization to build the subspace based on historical solutions. For these compared methods, we follow the original paper and set the SGD pretrained epoch as 50, after which we train for another 20 epochs using these methods. In total, the training epochs for all the compared methods and our proposed method are set to 70 to ensure a fair comparison. As shown in Tab. 2, although these methods operate within significantly reduced training dimensions, they do not perform well in terms of performance. Both P-SGD and TWA result in loss divergence, ultimately causing the training to crash. P-BFGS also struggles to converge, leading to a substantially higher R-D loss compared to our proposed method, 0.4057 vs 0.3550. Several factors contribute to these outcomes: (1) These methods were originally designed for image classification tasks and typically use small, simple datasets such as CIFAR10 and CIFAR100, whereas LIC training is inherently

---

[1]Adam is essentially SGD with first-order moments (i.e., mean) and second-order moments (i.e., variance) estimation.

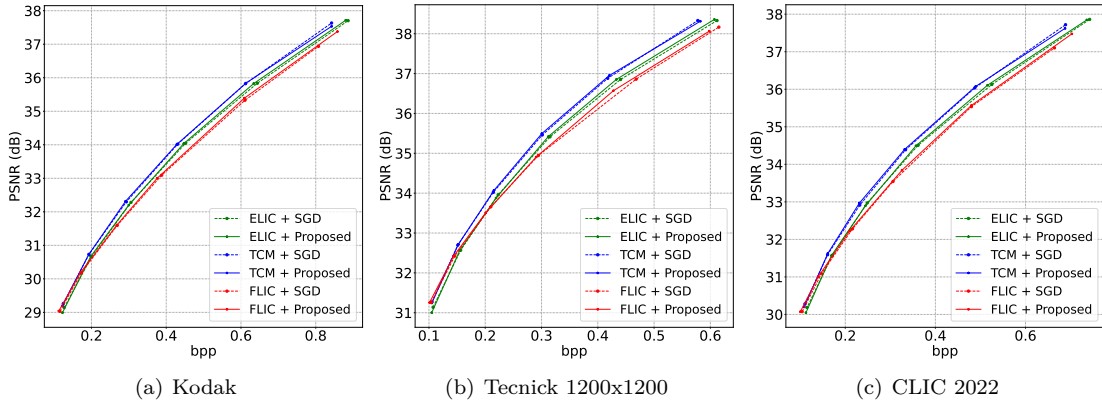

Figure 4: **R-D curves of various methods.** *Please zoom in for more details.*

more complex; (2) The dimensionality reduction in these methods heavily depends on historical solutions (sampled epochs), limiting the dimensionality to around 50. Directly projecting extremely complex networks into such small dimensions appears to cause non-convergence. Additionally, we observe that P-BFGS does not save training time compared to other efficient training methods, as the quasi-Newton techniques involve time-consuming estimation and iterative updates of the Hessian matrix.

Table 2: Comparison with Various Efficient Training Methods

| Settings | Training epoch ↓ | Training space dimension ↓ | Total training time ↓ | R-D loss ↓ |
|---|---|---|---|---|
| ELIC + SGD | 120 | 35,424,505 | 38h | 0.3544 |
| ELIC + P-SGD (Li et al., 2022a) | **70** | **40** | **23h** | div. |
| ELIC + P-BFGS (Li et al., 2022a) | **70** | **40** | 35h | 0.4057 |
| ELIC + TWA (Li et al., 2023a) | **70** | 50 | **23h** | div. |
| ELIC + Proposed | **70** | 50* | **23h** | **0.3550** |

**Train Conditions**:1 × Nvidia 4090 GPU, i9-14900K CPU, 128GB RAM. "div." indicates that these methods result in loss divergence, eventually causing the training to crash. *: The training dimension of our proposed method continues to decrease as training proceeds, theoretically converging to 50. $\lambda = 0.0018$. R-D loss $= \lambda \cdot 255^2 \cdot \text{MSE} + \text{Bpp}$. **Bold** indicates the best.

## 4.3 Complexity

We evaluate the complexity of SGD and our proposed method by measuring total training time, total trainable parameters for the whole training process, and trainable parameters in the final epoch, as presented in Tab. 1. Total training time is defined as the time required to train all models for each $\lambda$ value on a single 4090 GPU. Compared to models trained using SGD, our proposed method is significantly more efficient, reducing the training time to approximately 62% of that required by SGD. This efficiency is mainly attributed to the faster convergence of our method, which achieves the desired performance with fewer epochs while maintaining comparable training time per epoch. For example, training all FLIC models using SGD takes 520 hours ( 21 days on a single 4090 GPU), which is unbearable. In contrast, our proposed method drastically reduces the training time to 320 hours (13 days). Similarly, the training time for TCM-S is reduced from 346 hours (14 days) to 213 hours (8 days), highlighting the clear advantages of our approach.

The total trainable parameters are calculated by multiplying the model's trainable parameters by the total number of training epochs across all $\lambda$ values. For example, in the case of "ELIC + SGD", the total trainable parameters are computed as $35.42M \times 520$ epochs, resulting in $18,418M$. For "ELIC + Proposed", using $\lambda = 0.0018$ as an example, the total trainable parameters from epoch 1 to 20 are $35.42M \times 20$ epochs. Starting from epoch 21, we begin embedding 1% of the parameters in each epoch. Thus, for epochs 21 to 70, the total trainable parameters are calculated as $100\% \times 35.42M + 99\% \times 35.42M + \cdots + 51\% \times 35.42M$. By performing similar calculations for all other $\lambda$ values, we find that the total trainable parameters for "ELIC + Proposed" are $9,519M$, approximately 51% of the total trainable parameters required by the SGD

method. Similarly, the total trainable parameters for FLIC and TCM-S methods are reduced from $36,899$M (FLIC) and $23,493$M (TCM-S) to $19,070$M (FLIC) and $12,142$M (TCM-S), respectively, demonstrating a substantial reduction. We also report the final trainable parameters, representing the number of trainable parameters after the final epoch. For instance, FLIC's trainable parameters are reduced from 70.96M to 41.39M. By reducing the number of trainable parameters, we can further accelerate training by utilizing sparse training methods (Zhou et al., 2021), which will be discussed in Section A.2.

## 4.4 Ablation study

We conduct comprehensive experiments to find the impact of various factors of the proposed method. All experiment is conducted on ELIC (He et al., 2022) with $\lambda = 0.0018$. The other settings are the same as the main experiments. In the R-D plane (bpp-PSNR figures), the upper left represents better results.

### 4.4.1 What is the impact of the factors in the CMD calculation procedure?

In our method, three key factors significantly impact the step 1 CMD calculation. The first key factor is the predefined epoch $F$, which dictates when we perform CMD and the embedding mechanism begins. According to Fig. 2, the most significant loss changes occur within the first 20 epochs; therefore, we set $F$ to 20 to ensure that embedding neither starts too early nor too late. Starting the embedding process too early could compromise the accuracy of computed reference parameters and sensitivity estimations due to ongoing significant changes, adversely affecting performance. Conversely, beginning too late may delay convergence acceleration and reduce the number of embedded parameters due to fewer remaining epochs. To explore the effects of different settings, we tested $F$ values of 10, 20, and 30, as reported in Fig. 5(a). The results demonstrate that an $F$ of 10 leads to significantly poor convergence, while an $F$ of 30 does not improve upon the outcomes achieved with $F$ set to 20, therefore, we set $F$ to 20.

Additionally, the number of modes $M$ is critical in CMD computations. Insufficient modes fail to capture the correlations between parameters accurately, detrimentally affecting performance. Conversely, too many modes overcomplicate the computation and significantly increase memory demands, particularly when calculating the correlations between each parameter and the reference parameters. We experiment with $M = 10, 50$, and 100, shown in Fig. 5(a), and set 50 modes which offer the best balance. 10 modes are inadequate, resulting in low performance, while 100 modes, although slightly better than 10 modes are still worse than 50 modes, potentially due to the constraints of the numbers of sampled trajectories. Using a large number of modes to represent a few trajectories is not effective, but both excessive modes and extensive trajectory sampling contradict our efficiency goals. Therefore, 50 modes are confirmed as the most effective number.

To address the computational challenge of calculating the large $N \times N$ correlation matrix when finding reference parameters, we sample $S$ trajectories to determine the reference trajectories. A comparison of sampling 1k, 5k, 10k, and 20k trajectories, detailed in Fig. 5(a), reveals that 10k trajectories yield the best performance. Sampling 1k trajectories leads to inferior outcomes due to insufficient coverage of the entire parameter set. Sampling 5k trajectories is comparable to 10k. Sampling 20k trajectories slightly underperforms, likely necessitating additional modes to be effective, which also increases complexity. Hence, 10k is selected as it offers a more comprehensive representation than 5k, balancing effectiveness and efficiency.

### 4.4.2 What is the impact of factors of the embedding mechanism?

Determining the parameter sensitivity is a crucial process in the embedding mechanism. As mentioned earlier, embedding certain parameters may significantly reduce the performance. Therefore, we add experiments that omit sensitivity estimation, and we observe a substantial decrease in the R-D performance, see Fig. 5(b) "w/o Sensitivity". This significant change supports our hypothesis that certain parameters are extremely sensitive and inappropriate embedding leads to degraded performance.

Other factors influencing the embedding mechanism include the embedding percentage, $P$, and the embedding period, $L$. The percentage $P$ dictates the proportion of parameters to be embedded, while $L$ determines the frequency of embedding over specified epochs. As demonstrated in Fig. 5(b), experiments reveal that embedding 1% per epoch (Ours, $P = 1\%$ & $L = 1$) and 1% every two epochs ($P = 1\%$ & $L = 2$) yield the

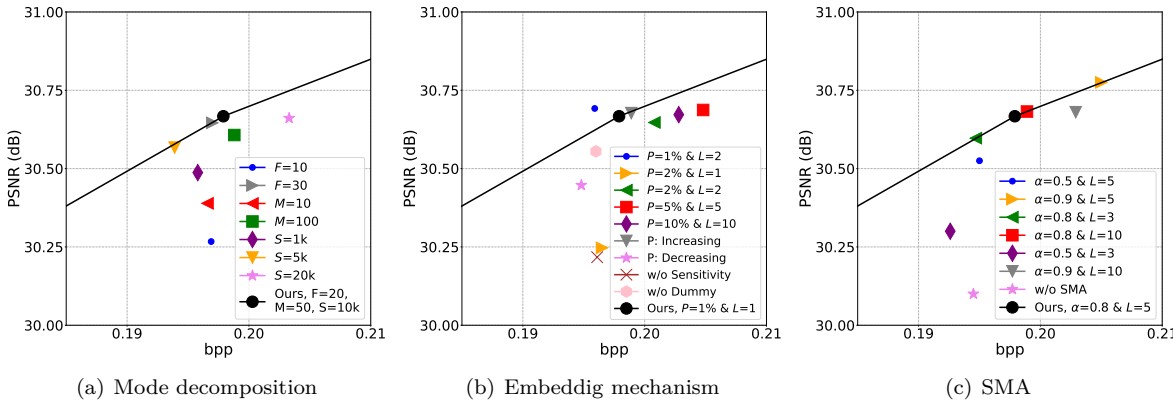

(a) Mode decomposition  (b) Embeddig mechanism  (c) SMA

Figure 5: **Ablation experiments on proposed methods.** ELIC model, $\lambda = 0.0018$.

best results. $P = 1\%$ & $L = 2$, however, results in too few parameters being embedded. Embedding 2% in a single epoch ($P = 2\%$ & $L = 1$) leads to poorer performance due to excessive embedding in the training process, resulting in inaccurate modeling and a decreased R-D performance. Comparing short-term small embedding percentages ($P = 1\%$ & $L = 1$ and $P = 2\%$ & $L = 2$) with long-term large ones ($P = 5\%$ & $L = 5$ and $P = 10\%$ & $L = 10$), we find that shorter smaller setup performs better. This likely stems from the disruptive effect of embedding a large number of parameters simultaneously, which adversely impacts training. Further experiments involving either a linear increase or a decrease scheduling $P$ show that gradually increasing the embedding percentage yields a performance similar to $P = 1\%$, $L = 1$, but results in more total trainable parameters. Conversely, a gradual decrease leads to poor performance. Therefore, we choose the straightforward approach of $P = 1\%$, $L = 1$ due to its simplicity and effectiveness.

The proposed dummy embedding involves reinitializing the original parameters using the embedded version during the embedding process, akin to the RWP technique, which has proven effective in enhancing performance. To evaluate the efficacy of dummy embedding, we conducted experiments in its absence. The results, detailed in Fig. 5(b) "w/o Dummy", indicate a performance drop upon removal of dummy embedding, thus affirming its positive impact on performance.

### 4.4.3   What is the impact of factors of the SMA?

We also conduct extensive experiments to assess the two critically important factors of SMA, as detailed in Fig. 5(c). A setting of $\alpha = 0.5, l = 5$ results in poorer performance, possibly due to an over-reliance on past states. In contrast, the setting of $\alpha = 0.9, l = 5$ shows no significant difference compared with utilized $\alpha = 0.8, l = 5$. When we adjust $l$, the values 3, 5, and 10 yield consistent performance. In our next experiments with new combinations of $\alpha$ and $l$, we find that $\alpha = 0.5, l = 3$ performs the worst. $\alpha = 0.9, l = 10$ is slightly less effective than $\alpha = 0.8, l = 5$. We then choose the setting $\alpha = 0.8, l = 5$ based on these results. Finally, we also attempted to remove the SMA, "w/o SMA". As previously discussed, the SMA is crucial for ensuring that the STDET operates smoothly. It is evident that the absence of the SMA results in a noticeable performance decline.

## 5   Conclusion

In this paper, we accelerate the training of LICs, by considering the linear representations of LICs' complex models. We extend the concept of Dynamic Mode Decomposition into our proposed Sensitivity-aware True and Dummy Embedding Training (STDET). Our method progressively embeds non-reference parameters based on their stable correlation and sensitivity, thereby reducing the dimension of the training space and the total number of trainable parameters. Additionally, we implement the Sampling-then-Moving Average (SMA) technique to smooth the training trajectory and minimize training variance, enhancing stable correlation. In this way, our approach results in faster convergence and fewer trainable parameters compared to standard SGD training of LICs. Both analyses on the noisy quadratic model and extensive experimental validations underscore the superiority of our method over standard training schemes for complex, time-intensive LICs.

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

## A  Appendix

This document provides more details about our proposed method and comparison.

### A.1  Experiments details

### A.1.1  Detailed Training Settings

We list detailed training information in Tab 3, including data augmentation, hyperparameters, and training devices. Please note that for all the experiments, we only use the default quantization methods (noise or STE) provided by the authors' implementations. We do not employ advanced quantization schemes to further fine-tune the models as stated in Sec. A.2.

### A.1.2  Used implementations

We list the implementations of the learning-based image codecs that we used for comparison in Tab. 4. We list the compared efficient training method implementations in Tab. 5

Table 3: Training Hyperparameters.

| | |
|---|---|
| Training set | COCO 2017 train |
| # images | 118,287 |
| Image size | Around 640x420 |
| Data augment. | Crop, h-flip |
| Train input size | 256x256 |
| Optimizer | Adam |
| Learning rate | $1 \times 10^{-4}$ |
| LR schedule | ReduceLROnPlateau |
| LR schedule parameters | factor: 0.5, patience: 5 |
| Batch size | 16 |
| Epochs (SGD) | 120 ($\lambda$ =0.0018), 80 (others) |
| Epochs (Proposed) | 70 ($\lambda$ =0.0018), 50 (others) |
| Gradient clip | 2.0 |
| GPUs | $1 \times$ RTX 4090 |
| STDET | |
| Predefined epoch $F$ | 20 ($\lambda$ =0.0018), 10 (others) |
| Sampled trajectories $S$ | 10000 |
| Modes $M$ | 50 |
| Embeddable parameters | 75% |
| Embedding period $L$ | 1 |
| True embedding percentage $P$ | 1% |
| Dummy embedding percentage | $\frac{t}{2}$%, $t$ is the current epoch |
| SMA | |
| Moving average factors $\alpha$ | 0.8 |
| Optimizer step $l$ | 5 |

Table 4: Learning-based Codecs Implementations.

| Method | Implementation |
|---|---|
| ELIC (He et al., 2022) | `https://github.com/InterDigitalInc/CompressAI` |
| TCM-S (Liu et al., 2023) | `https://github.com/jmliu206/LIC_TCM` |
| FLIC (Li et al., 2024a) | `https://github.com/qingshi9974/ICLR2024-FTIC` |

### A.2 Limitations and future work.

**Additional Memory Consumption:** The proposed method requires more memory due to additional non-learnable parameters. The notable additional parameters and their relative sizes are detailed below:

- STDET: Vectors $K$ and $D$, each with a size equal to that of the model parameters.

- STDET: A vector storing the mode related to each weight, with the same length as $K$ but containing positive integers representing mode numbers. This vector's size is approximately 0.25 times that of the model.

- SMA: SMA model parameters $w_{\text{SMA}}$, with a size equal to that of the model parameters.

Note that the memory used to store these parameters is insignificant during the training process compared to the sustained gradients (Izmailov et al., 2018), making the overall increase negligible.

**Overlook of Quantization:** As mentioned in A.1.1, we only used the default quantization methods (noise or STE) provided by the authors' implementations. According to Tsubota & Aizawa (2021), quantization is a fundamental operation in image compression and significantly affects the performance of LICs. In our

Table 5: Efficient Training Methods Implementations.

| Method | Implementation |
|--------|----------------|
| P-SGD (Li et al., 2022a) | `https://github.com/nblt/DLDR` |
| P-BFGS (Li et al., 2022a) | `https://github.com/nblt/DLDR` |
| TWA (Li et al., 2023a) | `https://github.com/nblt/TWA` |

experiment, we did not further fine-tune the models using the various quantization techniques outlined in the original papers, which may account for some discrepancies in the results compared to the original settings. Note, we ensured that all experiments were conducted under fair comparison conditions.

**Specific Domain Image Compression Acceleration:** In this paper, our focus is solely on natural image compression. We do not extend our scope to general coding schemes or specialized scenarios like content-adaptive coding and medical image compression. Each of these domains has unique requirements and challenges, such as the need for privacy considerations in medical imaging or the overfitting nature of content-adaptive coding. Addressing these scenarios would require tailored solutions beyond the scope of our current work. Our aim here is to improve the performance of natural image compression, which serves as a foundation that could inspire further research in these more specialized areas. We leave the acceleration of these methods for future work.

**Integration of Sparse Training:** With the embedding of LICs' parameters, they cease to be learnable, meaning that in practice, we can avoid the time-consuming gradient computations during backpropagation for these parameters. However, due to the constraints of existing platforms like TensorFlow and PyTorch, this is not straightforwardly achievable. These platforms rely on the chain rule to compute gradients, and simply setting "requires grad" to "False" does not prevent the computation of these parameters within the chain rule; it only avoids the storage of their gradients. A potential solution could involve integrating the proposed approach with sparse training techniques (Zhou et al., 2021), thereby completely bypassing gradient computations for these parameters during each training iteration. As a result, with an increasing number of embedded parameters, the training time per epoch would decrease, further accelerating the overall training process. However, this is not simple and beyond the scope of this paper, which we leave for future work.

### A.3 Variance reduction

We performed additional experiments to evaluate the variance in model performance across five training runs, comparing the average value and variance of performance between our method and SGD using ELIC (He et al., 2022) as a reference. Specifically, we trained the ELIC model with $\lambda = 0.0018$ five times using both our method and SGD. As shown in Tab. 6, our method achieves both lower average values and reduced variance compared to SGD. These results are consistent with our theoretical analysis, further underscoring the stability of the proposed method.

Table 6: Performance comparison across five runs.

| Method | Avg RD Loss (Variance) |
|--------|------------------------|
| ELIC + SGD | 0.3548 (9.04e-08) |
| ELIC + Proposed | 0.3543 (2.24e-08) |

### A.4 Discussion on STDET and SMA

To better understand the distinct roles of each proposed technique, we conducted additional experiments to evaluate them independently. All experimental settings remain the same as in the main experiments. We use the ELIC model (He et al., 2022) with $\lambda = 0.0018$ as the baseline method. The results are presented in Fig. 6.

First, we evaluated STDET without incorporating SMA (ELIC + STDET). When used on its own, STDET struggled to maintain competitive performance. This limitation stems from the stochastic nature of the training process, which introduces variability, affecting the consistency and accuracy of STDET's modeling. As a result, inaccurate correlation during parameter embedding leads to a noticeable performance drop.

Next, we applied SMA to a standard SGD-trained LIC model (ELIC + SMA). The results show that SMA achieves RD performance comparable to both standard SGD and the "STDET + SMA" setup, but it does not accelerate training. This suggests that SMA primarily functions as a smoothing technique and, when used alone, does not provide additional performance gains.

Thus, we observe the combination of STDET and SMA offers the best balance between training efficiency and performance. STDET reduces the training dimensionality and accelerates that training process, while SMA ensures stability throughout the process to enable accurate correlation modeling. Together, they enable an efficient and accurate acceleration of the LIC training process.

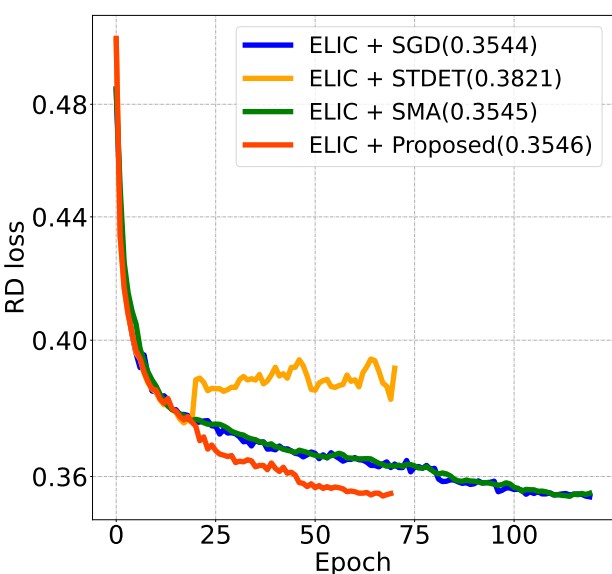

Figure 6: Testing loss comparison of various techniques. The upper right corner shows the final convergence result. $\lambda = 0.0018$, Testing RD loss $= \lambda \cdot 255^2 \cdot \text{MSE} + \text{Bpp}$. Evaluated on Kodak dataset.

### A.5 Detailed algorithm

The full practical STDET algorithm is presented in Algorithm. 1.

### A.6 Noisy quadratic analysis

In this section, we analyze the proposed method on a noisy quadratic model. The quadratic noise function is a commonly adopted base model for analyzing the optimization process, where the stochasticity incorporates noise introduced by mini-batch sampling (Schaul et al., 2013; Wu et al., 2018; Zhang et al., 2019b; Koh & Liang, 2017). We will show that the proposed method converges to a smaller steady-state training variance than the normal SGD training.

**Model Definition.** The quadratic noise model is defined as:

$$\hat{L}(\theta) = \frac{1}{2}(\theta - \mathbf{c})^T \mathbf{A}(\theta - \mathbf{c}), \tag{11}$$

where $\mathbf{c} \sim \mathcal{N}(\theta^*, \Sigma)$, $\mathbf{A}$ and $\Sigma$ are diagonal matrices, and $\theta^* = 0$ (Wu et al., 2018; Zhang et al., 2019a; 2023). Let $a_i$ and $\sigma_i^2$ be the $i$-th elements on the diagonal of $\mathbf{A}$ and $\Sigma$, respectively. The expected loss of

the iterates $\theta_t$ can be written as:

$$\mathcal{L}(\theta_t) = \mathbb{E}[\hat{L}(\theta_t)] = \frac{1}{2}\sum_i a_i(\mathbb{E}[\theta_{t,i}]^2 + \mathbb{V}[\theta_{t,i}] + \sigma_i^2), \tag{12}$$

where $\theta_{t,i}$ represents the $i$-th element of the parameter $\theta_t$. The steady-state risk of SGD and the proposed method are compared by unwrapping $\mathbb{E}[\theta_t]$ and $\mathbb{V}[\theta_t]$ in Eq. 12. The expectation trajectories $\mathbb{E}[\theta_t]$ contract to zero in both SGD and the proposed method. Thus, we focus on the variance:

**Steady-state risk.** Let $0 < \gamma < 1/L$ be the learning rate satisfying $L = \max_i a_i$. In the noisy quadratic setup, the variance of the iterates obtained by SGD and the proposed method converges to the following matrices:

$$V_{\text{SGD}}^* = \frac{\gamma^2 \mathbf{A}^2 \Sigma^2}{\mathbf{I} - (\mathbf{I} - \gamma\mathbf{A})^2}, \tag{13}$$

$$V_{\text{Proposed}}^* = \underbrace{\frac{(1 - p + p\mathbb{E}[k_i^2])\alpha^2(\mathbf{I} - (\mathbf{I} - \gamma\mathbf{A})^{2l})}{\alpha^2(\mathbf{I} - (\mathbf{I} - \gamma\mathbf{A})^{2l}) + 2\alpha(1 - \alpha)(\mathbf{I} - (\mathbf{I} - \gamma\mathbf{A})^l)}}_{\leq \mathbf{I}, \text{ if } \alpha \in (0,1) \text{ and } \mathbb{E}[k_i^2] \leq 1} V_{\text{SGD}}^*, \tag{14}$$

where $\alpha$ denotes the SMA weighting factor with varying trajectory states, $p$ is the percentage of final embedded parameters, and $\mathbb{E}[k_i^2]$ is the expectation of squared affine coefficients $k_i^2$.

From Eq. 14, it is evident that for the steady-state variance of the proposed method, the first product term is always smaller than $\mathbf{I}$ for $\alpha \in (0,1)$ and $\mathbb{E}[k_i^2] \leq 1$. Therefore, the proposed method exhibits a smaller steady-state variance than the SGD optimizer at the same learning rate, resulting in a lower expected loss (see Eq. 12). Additionally, due to the reduced variance in the training process, the correlation within the same modes is highly preserved, ensuring that the STDET avoids serious failure cases. The small steady-state variance ensures good generalizability in real-world scenarios (Wu et al., 2018) and demonstrates why the proposed method converges well.

***Proof.*** Here we present the proof of Eq. 14. We use $\theta$ to represent normal SGD training parameters and $\phi$ to represent SMA updated training parameters.

**Stochastic dynamics of SGD:** From Wu et al. (2018), we can obtain the dynamics of SGD with learning rate $\gamma$ as follows:

$$\mathbb{E}[\mathbf{x}^{(t+1)}] = (\mathbf{I} - \gamma\mathbf{A})\mathbb{E}[\mathbf{x}^{(t)}], \tag{15}$$

$$\mathbb{V}[\mathbf{x}^{(t+1)}] = (\mathbf{I} - \gamma\mathbf{A})^2\mathbb{V}[\mathbf{x}^{(t)}] + \gamma^2\mathbf{A}^2\Sigma. \tag{16}$$

**Stochastic dynamics of Proposed method:** We now compute the dynamics of the proposed method. The expectation and variance of the SMA have the following iterates based on the lookahead optmizers (Zhang et al., 2019b; 2023):

The expectation trajectory is represented as:

$$\begin{aligned}
\mathbb{E}[\phi_{t+1}] &= (1 - \alpha)\mathbb{E}[\phi_t] + \alpha\mathbb{E}[\theta_{t,l}] \\
&= (1 - \alpha)\mathbb{E}[\phi_t] + \alpha(\mathbf{I} - \gamma\mathbf{A})^l\mathbb{E}[\phi_t] \\
&= \left[1 - \alpha + \alpha(\mathbf{I} - \gamma\mathbf{A})^l\right]\mathbb{E}[\phi_t].
\end{aligned} \tag{17}$$

When combined with STDET, in the training dynamics, although the embedded parameters are not trainable, they depend on the reference parameters which are also updated by SMA. Thus all the parameters follow the same expectation dynamics:

For the variance, we can write

$$\mathbb{V}[\phi_{t+1}] = (1 - \alpha)\mathbb{V}[\phi_t] + \alpha^2\mathbb{V}[\theta_{t,l}] + 2\alpha(1 - \alpha)\text{cov}(\phi_t, \theta_{t,l}). \tag{18}$$

Also, it's easy to show:

$$\mathbb{V}[\theta_{t,l}] = (\mathbf{I} - \gamma\mathbf{A})^{2l}\mathbb{V}[\phi_t] + \gamma^2 \sum_{i=0}^{l-1}(\mathbf{I} - \gamma\mathbf{A})^{2i}\mathbf{A}^2\mathbf{\Sigma},$$

$$\text{cov}(\phi_t, \theta_{t,l}) = (\mathbf{I} - \gamma\mathbf{A})^l\mathbb{V}[\phi_t].$$

(19)

After substituting the SGD variance formula and some rearranging we have:

$$\mathbb{V}[\phi_{t+1}] = \left[1 - \alpha + \alpha(\mathbf{I} - \gamma\mathbf{A})^l\right]^2\mathbb{V}[\phi_t] + \alpha^2\sum_{i=0}^{l-1}(\mathbf{I} - \gamma\mathbf{A})^{2i}\gamma^2\mathbf{A}^2\mathbf{\Sigma}.$$

(20)

Similarly, for the variance term, all the parameters follow the same SMA dynamics. Then for the proposed method:

$$\mathbb{E}[\phi_{t+1}] = \left[1 - \alpha + \alpha(\mathbf{I} - \gamma\mathbf{A})^l\right]\mathbb{E}[\phi_t],$$

(21)

$$\mathbb{V}[\phi_{t+1}] = \left[1 - \alpha + \alpha(\mathbf{I} - \gamma\mathbf{A})^l\right]^2\mathbb{V}[\phi_t] + \alpha^2\sum_{i=0}^{l-1}(\mathbf{I} - \gamma\mathbf{A})^{2i}\gamma^2\mathbf{A}^2\mathbf{\Sigma}.$$

(22)

Now, we proceed to find the steady state of the expectation and variance:

The given expectation term is:

$$\mathbb{E}[\phi_{t+1}] = \left[1 - \alpha + \alpha(\mathbf{I} - \gamma\mathbf{A})^l\right]\mathbb{E}[\phi_t].$$

(23)

Define the mapping $T$ as:

$$T(\phi_t) = \left[1 - \alpha + \alpha(\mathbf{I} - \gamma\mathbf{A})^l\right]\phi_t.$$

(24)

A mapping $T$ is a contraction if there exists a constant $0 \le c < 1$ such that for all $\phi_t$ and $\phi_t'$:

$$\|T(\phi_t) - T(\phi_t')\| \le c\|\phi_t - \phi_t'\|.$$

(25)

In our case:

$$\|T(\phi_t) - T(\phi_t')\| = \left\|\left[1 - \alpha + \alpha(\mathbf{I} - \gamma\mathbf{A})^l\right]\phi_t - \left[1 - \alpha + \alpha(\mathbf{I} - \gamma\mathbf{A})^l\right]\phi_t'\right\|$$
$$= \left\|\left[1 - \alpha + \alpha(\mathbf{I} - \gamma\mathbf{A})^l\right](\phi_t - \phi_t')\right\|.$$

(26)

Let $M = 1 - \alpha + \alpha(\mathbf{I} - \gamma\mathbf{A})^l$. We need to find the spectral radius $\rho(M)$, which represents the largest absolute value of the eigenvalues of $M$. Since $(\mathbf{I} - \gamma\mathbf{A})$ is a contraction matrix (assuming the learning rate $\gamma$ is chosen such that $0 < \gamma < 1/L$, where $L = \max_i a_i$), its eigenvalues $\lambda_i$ satisfy:

$$|\lambda_i| < 1.$$

(27)

Therefore, the eigenvalues of $(\mathbf{I} - \gamma\mathbf{A})^l$ are $\lambda_i^l$ and satisfy:

$$|\lambda_i^l| < 1.$$

(28)

Next, consider the matrix $M$:

$$M = 1 - \alpha + \alpha(\mathbf{I} - \gamma\mathbf{A})^l.$$

(29)

The eigenvalues of $M$ are given by:

$$\rho(M) = \left|1 - \alpha + \alpha\lambda_i^l\right|.$$

(30)

For the mapping $T$ to be a contraction, the spectral radius $\rho(M)$ must be less than 1:

$$\left|1 - \alpha + \alpha\lambda_i^l\right| < 1$$

(31)

Given $0 < \alpha < 1$ and $|\lambda_i^l| < 1$, the inequality $\left|1 - \alpha + \alpha\lambda_i^l\right| < 1$ will hold. By Banach's Fixed Point Theorem (Oltra & Valero, 2004), since $T$ is a contraction mapping, there exists a unique fixed point $E^*$ such that $T(E^*) = E^*$.

To find the fixed point, we set $\mathbb{E}[\phi_{t+1}] = \mathbb{E}[\phi_t] = E^*$:

$$E^* = \left[1 - \alpha + \alpha(\mathbf{I} - \gamma\mathbf{A})^l\right] E^*. \tag{32}$$

Simplifying:

$$\left[\alpha - \alpha(\mathbf{I} - \gamma\mathbf{A})^l\right] E^* = 0. \tag{33}$$

Since $\alpha \neq 0$. The only solution is:

$$E^* = 0. \tag{34}$$

Using the contraction mapping approach, we have shown that the expectation term indeed converges to the fixed point $E^* = 0$.

Now, we need to further consider the effect of STDET. Let $p$ represent the percentage of final embedded parameters. Thus $E^* = pE^*_{\text{embed}} + (1-p)E^*_{\neg\text{embed}}$, where $E^*_{\text{embed}} \to 0$ and $E^*_{\neg\text{embed}} = \mathbb{E}[k_i]E^*_{\text{embed}} + \mathbb{E}[d_i] \to \mathbb{E}[d_i]$, where we empirically find that $\mathbb{E}[d_i] \to 0$ in our cases. Thus overall the expectation trajectories contract to zero. Clearly, the expectation trajectories of SGD also contract to zero as it is also a contract map.

Similarly, for the variance, under the same condition, we know that:

$$\begin{aligned} V^*_{\text{SGD}} &= (1 - \gamma\mathbf{A})^2 V^*_{\text{SGD}} + \gamma^2\mathbf{A}^2\Sigma \\ &= \frac{\gamma^2\mathbf{A}^2\Sigma}{\mathbf{I} - (\mathbf{I} - \gamma\mathbf{A})^2}. \end{aligned} \tag{35}$$

For the proposed method,

$$\begin{aligned} V^*_{\text{SMA}} &= \left[1 - \alpha + \alpha(\mathbf{I} - \gamma\mathbf{A})^l\right]^2 V^*_{\text{SMA}} + \alpha^2\sum_{i=0}^{l-1}(\mathbf{I} - \gamma\mathbf{A})^{2i}\gamma^2\mathbf{A}^2\Sigma \\ &= \frac{\alpha^2\sum_{i=0}^{l-1}(\mathbf{I} - \gamma\mathbf{A})^{2i}}{\mathbf{I} - [(1-\alpha)\mathbf{I} + \alpha(\mathbf{I} - \gamma\mathbf{A})^l]^2}\gamma^2\mathbf{A}^2\Sigma \\ &= \frac{\alpha^2(\mathbf{I} - (\mathbf{I} - \gamma\mathbf{A})^{2l})}{\mathbf{I} - [(1-\alpha)\mathbf{I} + \alpha(\mathbf{I} - \gamma\mathbf{A})^l]^2}\frac{\gamma^2\mathbf{A}^2\Sigma}{\mathbf{I} - (\mathbf{I} - \gamma\mathbf{A})^2} \\ &= \frac{\alpha^2(\mathbf{I} - (\mathbf{I} - \gamma\mathbf{A})^{2l})}{\alpha^2(\mathbf{I} - (\mathbf{I} - \gamma\mathbf{A})^{2l}) + 2\alpha(1-\alpha)(\mathbf{I} - (\mathbf{I} - \gamma\mathbf{A})^l)}\frac{\gamma^2\mathbf{A}^2\Sigma^2}{\mathbf{I} - (\mathbf{I} - \gamma\mathbf{A})^2} \\ &= \frac{\alpha^2(\mathbf{I} - (\mathbf{I} - \gamma\mathbf{A})^{2l})}{\alpha^2(\mathbf{I} - (\mathbf{I} - \gamma\mathbf{A})^{2l}) + 2\alpha(1-\alpha)(\mathbf{I} - (\mathbf{I} - \gamma\mathbf{A})^l)}V^*_{\text{SGD}}. \end{aligned} \tag{36}$$

Also, together with the STDET:

$$\begin{aligned} V^*_{\text{Proposed}} &= pV^*_{\text{embed}} + (1-p)V^*_{\neg\text{embed}} \\ &= p\mathbb{E}[k_i^2])V^*_{\text{SMA}} + (1-p)V^*_{\text{SMA}} \\ &= \underbrace{\frac{(1 - p + p\mathbb{E}[k_i^2])\alpha^2(\mathbf{I} - (\mathbf{I} - \gamma\mathbf{A})^{2l})}{\alpha^2(\mathbf{I} - (\mathbf{I} - \gamma\mathbf{A})^{2l}) + 2\alpha(1-\alpha)(\mathbf{I} - (\mathbf{I} - \gamma\mathbf{A})^l)}}_{\leq\mathbf{I},\ \text{if}\ \alpha\in(0,1)\ \text{and}\ \mathbb{E}[k_i^2]\leq 1}V^*_{\text{SGD}}, \end{aligned} \tag{37}$$

Thus, we have shown Eq. 14 and the proposed method will achieve a smaller loss for the same learning rate as the variance is reduced in Eq. 12.

---

**Algorithm 1** STDET Algorithm

---

**Hyper-parameters:**

- $F$ (Predefined epochs)

- $L$ (Embedding period)

- $P$ (Embedding percentage)

- Other inputs required for the training process and STDET (learning rate, etc.)

**Procedure:**

1. Run $F$ regular SGD epochs to obtain weight trajectories $W_F$.

2. Perform CMD to select the reference trajectory $w_{m,F}$, determine the related modes and compute affine coefficients $K$ and $D$ .

3. Find embeddable parameters by calculating the parameter sensitivity.

4. Initialize the set of embedded weights: $\mathcal{I} \leftarrow \emptyset$.

5. **for** $t > F$ **do**

   (a) Perform a regular SGD epoch.
   (b) Iteratively update $K$ and $D$ using Eq. 6.
   (c) Update weights $w_i(t)$ as follows:

   $$w_i(t) \leftarrow \begin{cases} k_i w_r(t) + d_i & \text{if } i \in \mathcal{I} \\ \text{SGD update} & \text{if } i \notin \mathcal{I} \end{cases}$$

   (d) **if** $(t - F)\%L == 0$ **then**
       i. Compute the long-term change:

       $$C(t) = \sqrt{(K(t) - K(t - L))^2 + (D(t) - D(t - L))^2}$$

       ii. Update $\mathcal{I}$ to include the weights with the least $P\%$ changes in terms of $C$, excluding reference weights and already embedded weights.
       iii. Reinitialize additional $\frac{tP}{2}\%$ parameters that have the least $C$ changes excluding $\mathcal{I}$ by: $w_i \leftarrow k_i w_m(t) + d_i$.
   (e) **end if**

6. **end for**

---

