# OpenReview forum: "Accelerating Learned Image Compression with Sensitivity-aware Embedding and Moving Average"
_TMLR — Rejected by TMLR_

### Review · Reviewer_LALm · 2024-10-04

**Summary Of Contributions:**

The paper aims to accelerate the convergence of LIC models, reducing both the number of trainable parameters and training time, without sacrificing model performance. The two core techniques are Sensitivity-aware True and Dummy Embedding Training (STDET) and Sampling-then-Moving Average (SMA).

**Audience:**

Yes

**Claims And Evidence:**

Yes

**Requested Changes:**

Please look at the weakness. It is necessary to clarify the performance issue of FLIC and how to efficiently select appropriate hyper - parameters.

**Strengths And Weaknesses:**

Strength:
1. The paper attempts to accelerate the training process of learned image compression, which is very important for the current situation where the training time required for image compression is gradually increasing.
2. The paper has rigorous theoretical derivations and clear problem analysis.

Weakness:
1. In Figure 4, the performance of FLIC is different from that in the original paper. In the original paper, the performance is better than that of TCM and ELIC, but in the figure, it shows that it is the method with the worst performance.
2. Figure 5 shows that the adjustment of hyper - parameters has a great influence on the final performance. How to select the most appropriate parameters for different models?

---

> ### Author Response · Authors · 2024-10-11
> **Response to Reviewer LALm**
>
> We are grateful to the reviewer for the thoughtful comments and suggestions. Each point has been thoroughly addressed, and we have made the corresponding adjustments to the manuscript, as outlined in the following responses.
>
> **1. Performance of FLIC:** Thank you for pointing this out. We used the author-provided code available at: https://github.com/qingshi9974/ICLR2024-FTIC and followed the GitHub instructions to reproduce the results. However, our experimental setup differed slightly from the original, leading to the observed discrepancies:
> 1. **Training Dataset:** For a fair comparison across all methods, we used the COCO2017 dataset for training. In contrast, the original paper trained on Flickr2W and ImageNet-1k, which could account for the performance differences.
>
> 2. **Quantization Implementation:** As mentioned in our paper (Section A.2: Limitations and Future Work), we used the default quantization methods (either noise-based or Straight-Through Estimator (STE)) provided in the authors' implementation. According to Tsubota & Aizawa (2021), quantization plays a critical role in image compression performance. However, we did not further fine-tune these models with the various quantization techniques described in their paper. Specifically, in the provided code and instruction, the authors applied `STE(z - z offset) + z offset` and `STE(y)`, and we followed this setting to train the model end-to-end. In the original paper, however, they employed a more sophisticated training strategy: during the first stage, they trained models using only a hyperprior as the entropy model to develop strong non-linear transforms, applying `STE(z - z offset) + z offset` and `STE(y - μ) + μ`. In the second stage, they fine-tuned the transforms using the checkpoint from the first stage and jointly trained the transforms with a randomly initialized T-CA, applying `STE(z - z offset) + z offset` and `STE(y)` due to the causality constraint during this stage.
>
> These differences in the training process and quantization techniques likely explain the performance variation. However, we ensured that all experiments were conducted under fair comparison conditions.  We further clarified this point in the revised manuscript (Appendix A.2).
>
> **2. Select the most appropriate parameters:** We appreciate the reviewer's insightful comment regarding hyperparameter sensitivity.
> To select the best hyperparameters for each different model, one could do a brute-force hyperparameter search (which can be computationally expensive).
> Fortunately, a well-determined set of hyperparameters can serve as a general configuration that works well in most cases. In our experiments, we applied the same set of hyperparameters across all LIC methods, and this yielded strong performance across different models. This suggests that our method is fairly robust and can achieve good results without requiring extensive hyperparameter tuning for each model.

---

### Review · Reviewer_6hAH · 2024-10-06

**Summary Of Contributions:**

The paper focuses on enhancing the efficiency of training Learned Image Compression (LIC) models, two specific techniques are introduced to reduce the computational burden without sacrificing performance:

1. Sensitivity-aware True and Dummy Embedding Training (STDET) to model the LIC training process by identifying parameter sensitivity and leveraging intra-mode correlations among parameters.  It selectively embeds non-reference parameters that demonstrate stable correlations with reference parameters in the same mode. By embedding these stable parameters early in the training, the model reduces the number of active trainable parameters over time, allowing the focus to shift to more sensitive, non-embedded parameters.

2. Sampling-then-Moving Average (SMA) to smooth the training trajectory by averaging parameter updates. It samples parameter states during training and applies a moving average to interpolate the weights, thus ensuring more stable updates and reducing variance in the final model parameters.

The paper also includes a theoretical analysis using the noisy quadratic model, demonstrating that the proposed methods achieve lower training variances than standard SGD.

**Audience:**

Yes

**Claims And Evidence:**

Yes

**Requested Changes:**

See Cons above

**Strengths And Weaknesses:**

Pros:
1. The proposed approach offers a significant reduction in training time, This speedup is especially noteworthy in complex models like ELIC, TCM-S, and FLIC.

2. Despite the accelerated training, the method maintains comparable model performance, ensuring no trade-off between speed and quality.

3. The method is grounded in a rigorous theoretical framework, supported by analysis of the noisy quadratic model.

Cons:
1. I feel it lacks of motivation for training speed focus as it primarily focuses on improving training speed, which isn't typically the main concern in image compression. In most cases, we prioritize inference speed (i.e., encoding and decoding times), unless the goal is content-adaptive coding. The authors should consider including experiments related to content-adaptive coding to better demonstrate the advantages of their method in this context.

2. The performance of the proposed method heavily relies on specific hyperparameters, such as the number of modes and sampled trajectories. In real-world applications, optimal hyperparameter settings can vary depending on the domain—compressing natural images may require different settings than compressing medical images. This high sensitivity to hyperparameters could limit the practicality of the method for general coding tasks.

Therefore, while this paper offers interesting insights for Learned Image Compression (LIC), its practical deployment in the near future seems unlikely. The focus on training speed and high sensitivity to hyperparameters make it less suitable for immediate real-world applications, particularly when inference speed and robustness across different coding domains are more critical.

---

> ### Author Response · Authors · 2024-10-11
> **Response to Reviewer 6hAH**
>
> We sincerely thank the reviewer for the valuable insights and constructive feedback. We have carefully considered each of the comments provided and have made the necessary revisions to the manuscript, as detailed below.
>
> **1. Motivation for training speed:** We appreciate the reviewer’s thoughtful feedback on the motivation to improve the training speed. While we acknowledge that inference speed is often the key priority, we stress that training speed is also critical. For example, in codec development, one needs to design model architectures and search for hyperparameters to meet requirements on each separate data source, bit rate, computation complexity constraints, etc.
> This involves multiple training runs, and efficient training thus becomes especially important.
> We have revised the introduction (Section 1) to better motivate our research:
> “For instance, training the FLIC models takes up to 520 hours (21 days) using a single 4090 GPU (Table 1), while the design process of the FLIC models likely requires even more GPU resources, as it involves extensive experimentation, including hyper-parameter tuning, structural adjustments, and other iterative processes. If the training speed of these models is not improved, the time required to develop new methods may become prohibitive. Therefore, developing strategies to reduce training times is of paramount importance."
> We also want to mention that, as commented by Reviewer LALm: "The paper attempts to accelerate the training process of learned image compression, which is very important given the increasing time required for training image compression models." We thus believe that the training speed issue is not negligible and should be addressed promptly.
>
> We also recognize the importance of content-adaptive coding. However, the investigation of accelerating content-adaptive coding is beyond the scope of this paper, as it presents unique challenges which would require specialized approaches to address them. We do not anticipate that a single method can effectively address the acceleration needs for all specific compression tasks.
>
> **2. Limit the practicality of the method for general coding tasks:** We appreciate the reviewer's insightful comment on hyperparameter sensitivity. We acknowledge that hyperparameter tuning can indeed differ based on the problem domain. However, we would like to note that the same set of hyperparameters was used across all the LIC methods in our experiments, which indicates that strong performance can be achieved without extensive hyperparameter tuning for different methods. This suggests that the method is reasonably robust across various LICs without requiring significant adjustments to the hyperparameters.
>
> Furthermore, we agree that specific domains, such as medical image compression or content-adaptive coding, often require tailored approaches. Given the unique characteristics of these problems, it is unrealistic to expect a single natural image compression method to be effective across all scenarios. For example, in medical image compression [1,2,3], many methods are designed specifically to address aspects like privacy, 3D representation, or near-lossless compression, rather than being simple adaptations of existing natural image compression techniques [4,5,6]. Similarly, acceleration strategies may also need to be customized for each domain which represents an intriguing direction for future research. We have also included additional discussion about content-adaptive coding and medical image compression in Appendix A.2, under "Limitations and Future Work-Specific Domain Image Compression Acceleration."
>
> [1]. Li, Zhongqiang, et al. "Nearly-lossless-to-lossy medical image compression by the optimized JPEGXT and JPEG algorithms through the anatomical regions of interest." *Biomedical Signal Processing and Control* 83 (2023): 104711.
>
> [2]. Rossinelli, Diego, et al. "High-throughput lossy-to-lossless 3D image compression." *IEEE Transactions on Medical Imaging* 40.2 (2020): 607-620.
>
> [3]. Mishra, Dipti, Satish Kumar Singh, and Rajat Kumar Singh. "Deep architectures for image compression: a critical review." *Signal Processing* 191 (2022): 108346.
>
> [4]. He, Dailan, et al. "Elic: Efficient learned image compression with unevenly grouped space-channel contextual adaptive coding." *Proceedings of the IEEE/CVF Conference on Computer Vision and Pattern Recognition.* 2022.
>
> [5]. Liu, Jinming, Heming Sun, and Jiro Katto. "Learned image compression with mixed transformer-cnn architectures." *Proceedings of the IEEE/CVF Conference on Computer Vision and Pattern Recognition.* 2023.
>
> [6]. Li, Han, et al. "Frequency-Aware Transformer for Learned Image Compression." *The Twelfth International Conference on Learning Representations.*

---

### Review · Reviewer_FDaE · 2024-10-07

**Summary Of Contributions:**

This paper aims at accelerating the training of learned image compression methods by modeling the training dynamics. It proposes a Sensitivity-aware True and Dummy Embedding Training mechanism which clusters the LIC model parameters into a few separate modes. The reference parameters within the same mode are expressed as affine transformations. The non-reference parameters are gradually embedded, and the number of trainable parameters is reduced. A Sampling-then-Moving Average technique is proposed to interpolate sampled weights from SGD training to obtain the moving averaged weights to ensure smooth temporal behavior and minimize the training state variances.

**Audience:**

Yes

**Broader Impact Concerns:**

No concern.

**Claims And Evidence:**

Yes

**Requested Changes:**

See the weakness part

**Strengths And Weaknesses:**

## Strengths
- The proposed method reduces training space dimensions and the number of trainable parameters without sacrificing model performance.
- A theoretical analysis of the Noisy quadratic model is given to show that the proposed method achieves a lower training variance than standard SGD.

## Weaknesses
- Additional experiments. The experiment section has shown that the proposed training methods can accelerate training without sacrificing model performance. Could you give an extra figure to show the performance (y-axis) at different epochs (x-axis) when using SGD and your proposed methods?
- Another question is about the stability of the proposed method. Though the proposed method can accelerate training process, will it damage the training stability? Could you show the variance of the model performance to compare SGD and your proposed method?
- Performance at different bitrate. Because that the latent distribution will vary significantly at different rate point, which may also affect the training. Will the proposed method show different characteristics at low bitrate and high bitrate?

---

> ### Author Response · Authors · 2024-10-11
> **Response to Reviewer FDaE**
>
> We appreciate the insightful feedback provided by the reviewer. We have addressed each of the points raised and made the corresponding changes to the manuscript as outlined below.
>
> **1. Additional figures:** Thank you for the suggestion. We have included a performance versus epoch figure in the paper (see Figure 2: Testing loss comparison of various methods). This figure demonstrates that the proposed method converges significantly faster than standard SGD across different LICs. Additionally, as highlighted in the upper-right corner of the figure, our method reaches a similar final convergence level compared to SGD. The RD loss is defined as: $\text{RD loss} = \lambda \cdot 255^2 \cdot \text{MSE} + \text{Bpp}$.
>
> **2. Stability of the proposed method:** Thanks to the reviewer for the insightful comment.
> We agree that training stability is an important aspect to consider. In the revised paper, we add additional experiments that evaluate the variance of model performance across five training runs.
> The experiments compare the average value and variance of performance between our method and SGD. The results, presented in Appendix A.3 Table 6 of the revised manuscript, show that our method consistently demonstrates both lower average values and variance compared to SGD. This aligns with our theoretical analysis, further supporting the stability of our proposed method.
>
> **Table**: Performance comparison across five runs.
> | **Method**         | **Avg RD Loss (Variance)** |
> |--------------------|---------------------------|
> | ELIC + SGD         | 0.3548 (9.04e-08)         |
> | ELIC + Proposed    | 0.3543 (2.24e-08)         |
>
>
> **3. Performance at different bitrate:** We agree that the latent distribution varies at different bitrates. As demonstrated in Figure 4 (R-D curves of various methods), our method consistently performs well across a wide range of bitrates. The rate points of the models trained with SGD and our proposed method nearly overlap, indicating that our approach maintains stability and performance across different bitrate settings, without being negatively impacted by bitrate variations.

---

### Review · Reviewer_CB7R · 2024-10-14

**Summary Of Contributions:**

This manuscript proposes several techniques to significantly enhance the training efficiency of learned image codecs (LICs). By analyzing the clustered modes of LIC parameter trajectories, the authors introduce a Sensitivity-aware True and Dummy Embedding Training (STDET) mechanism. It reduces both the training space dimension and the number of trainable parameters, thereby lowering training complexity. Additionally, they propose a Sampling-then-Moving Average (SMA) method to improve training smoothness. Experimental results show a substantial reduction in training time.

**Audience:**

Yes

**Broader Impact Concerns:**

No concerns

**Claims And Evidence:**

Yes

**Requested Changes:**

Please refer to the weakness part.

**Strengths And Weaknesses:**

Strengths:

1. The experimental results demonstrate the effectiveness of the proposal in reducing training time without sacrificing compression performance.
2. The theoretical analysis using noisy quadratic model makes the results more conviable.

Weakness :
1. In Figure 1(b), some percentage values for the table are missing (e.g., loss %).
2. The effectiveness of STDET and SMA should be verified separately. Figure 5(c) shows the importance of SMA, with its removal causing a 0.5 dB drop in quality. This raises the question: does applying SMA to normally trained LICs (e.g., SGD-based) provide similar gains? If so, SMA could be a standalone technique for enhancing LICs, rather than solely complementing STDET for smoothing. In other words, if "STDET + SMA" does not outperform "SGD + SMA," the effectiveness of STDET alone remains unconvincing. Clarification is needed to help readers better understand the distinct roles of STDET and SMA.

---

> ### Author Response · Authors · 2024-10-15
> **Response to Reviewer CB7R**
>
> We would like to thank the reviewer for the insightful comments. Below, we address the points raised:
>
> **1. Missing "\%":** Thank you for highlighting this issue. We have revised the table to include the missing percentage values (e.g., loss \%) for completeness and clarity.
>
> **2. The effectiveness of STDET and SMA should be verified separately:** We appreciate the reviewer’s comment regarding the combined evaluation of STDET and SMA. In response, we have conducted additional experiments to assess each technique independently, as shown in Appendix A.4 "Discussion on STDET and SMA":
>
> 1. **Effectiveness of STDET:** We evaluated STDET without incorporating SMA (ELIC + STDET). When used on its own, STDET struggled to maintain competitive performance. This limitation stems from the stochastic nature of the training process, which introduces variability, affecting the consistency and accuracy of STDET’s modeling. As a result, inaccurate correlation during parameter embedding leads to a noticeable performance drop.
>
> 2. **Effectiveness of SMA:** We applied SMA to a standard SGD-trained LIC model (ELIC + SMA). The results show that SMA achieves RD performance comparable to both standard SGD and the "STDET + SMA" setup, but it does not accelerate training. This suggests that SMA primarily functions as a smoothing technique and, when used alone, does not provide additional performance gains.
>
> Thus, we observe the combination of STDET and SMA offers the best balance between training efficiency and performance. STDET reduces the training dimensionality and accelerates that training process, while SMA ensures stability throughout the process to enable accurate correlation modeling. Together, they enable an efficient and accurate acceleration of the LIC training process.

---

### Decision · Action_Editor_3Qa5 · 2024-11-20

**Recommendation:** Reject

**Comment:**

We received quite mixed recommendations from the reviewers, with two leaning towards acceptance and two leaning towards rejection. Overall, we acknowledge that the technical aspects of the research are strong. However, there is a lack of compelling evidence for its utility in real-world applications. Additionally, the authors should enhance the articulation of the research's practical impacts, demonstrate its adaptability across different settings and configurations, and consider the evolving nature of LIC architectures more thoroughly. The authors may consider submitting a major revision at a later time.

**Audience:**

The technical contributions of the research could appeal to those in the field who are focused on theoretical advancements and methodological innovations. The exploration of hyperparameter uniformity and adaptability to different image domains could provide valuable insights for researchers working on similar problems or those interested in the evolving architectures within Learning in Context (LIC).

**Claims And Evidence:**

Reviewers' concerns highlight a deficiency in articulating the practical relevance and broader impact of the research. There's also an insufficient demonstration of the method's adaptability across varied settings and a question regarding the uniformity of hyperparameters across different image domains. Additionally, the paper does not appear to adequately address the evolving architectures in the Learning in Context (LIC) field, casting doubts about the method's adaptability to new and diverse architectures.

**Resubmission Of Major Revision:**

The authors may consider submitting a major revision at a later time.